# Evaluating tubulointerstitial compartments in renal biopsy specimens using a deep learning-based approach for classifying normal and abnormal tubules

**Satoshi Hara[1,2☯], Emi Haneda[3☯], Masaki Kawakami[3], Kento Morita[3], Ryo Nishioka[2], Takeshi Zoshima[2], Mitsuhiro Kometani[4], Takashi Yoneda[4,5,6], Mitsuhiro Kawano[ID 2]\*, Shigehiro Karashima[7], Hidetaka Nambo[3]\***

1 Medical Education Research Center, Graduate School of Medical Sciences, Kanazawa University, Kanazawa, Japan, 2 Department of Rheumatology, Kanazawa University Graduate School of Medicine, Kanazawa, Japan, 3 School of Electrical Information Communication Engineering, College of Science and Engineering, Kanazawa University, Kanazawa, Japan, 4 Department of Endocrinology and Metabolism, Kanazawa University Graduate School of Medicine, Kanazawa, Japan, 5 Department of Health Promotion and Medicine of the Future, Kanazawa University Graduate School of Medicine, Kanazawa, Japan, 6 Faculty of Transdisciplinary Sciences, Institute of Transdisciplinary Sciences, Kanazawa University, Kanazawa, Japan, 7 Institute of Liberal Arts and Science, Kanazawa University, Kanazawa, Japan

☯ These authors contributed equally to this work.
\* sk33166@gmail.com (MK); nambo@blitz.ec.t.kanazawa-u.ac.jp (HN)

**Data Availability Statement:** All relevant data are within the paper and its Supporting Information files.

## Abstract

Renal pathology is essential for diagnosing and assessing the severity and prognosis of kidney diseases. Deep learning-based approaches have developed rapidly and have been applied in renal pathology. However, methods for the automated classification of normal and abnormal renal tubules remain scarce. Using a deep learning-based method, we aimed to classify normal and abnormal renal tubules, thereby assisting renal pathologists in the evaluation of renal biopsy specimens. Consequently, we developed a U-Net-based segmentation model using randomly selected regions obtained from 21 renal biopsy specimens. Further, we verified its performance in multiclass segmentation by calculating the Dice coefficients (DCs). We used 15 cases of tubulointerstitial nephritis to assess its applicability in aiding routine diagnoses conducted by renal pathologists and calculated the agreement ratio between diagnoses conducted by two renal pathologists and the time taken for evaluation. We also determined whether such diagnoses were improved when the output of segmentation was considered. The glomeruli and interstitium had the highest DCs, whereas the normal and abnormal renal tubules had intermediate DCs. Following the detailed evaluation of the tubulointerstitial compartments, the proximal, distal, atrophied, and degenerated tubules had intermediate DCs, whereas the arteries and inflamed tubules had low DCs. The annotation and output areas involving normal and abnormal tubules were strongly correlated in each class. The pathological concordance for the glomerular count, t, ct, and ci scores of the Banff classification of renal allograft pathology remained high with or without the segmented images. However, in terms of time consumption, the quantitative assessment of tubulitis, tubular atrophy, degenerated tubules, and the interstitium was

**Funding:** The authors received no specific funding for this work.

**Competing interests:** The authors have declared that no competing interests exist.

improved significantly when renal pathologists considered the segmentation output. Deep learning algorithms can assist renal pathologists in the classification of normal and abnormal tubules in renal biopsy specimens, thereby facilitating the enhancement of renal pathology and ensuring appropriate clinical decisions.

## Introduction

Renal diseases are a significant global burden in all facets of health and economy [1, 2]. Therefore, the precise diagnosis of kidney diseases is a prerequisite for selecting an appropriate treatment strategy. As the golden standard for diagnosing kidney diseases, renal pathology is essential. Information obtained from renal biopsy specimens is used to confirm the diagnosis and further assess the severity and prognosis of kidney disease. Therefore, to ensure appropriate clinical decisions, the accurate assessment of renal biopsy specimens is essential.

Currently, deep learning-based approaches have developed rapidly, and they have been applied extensively in the subspecialty of renal pathology [3]. Specifically, convolutional neural networks (CNNs), which are the most popular deep learning-based techniques, are mainly used for the automated detection and morphometric analysis of histological components and in the prediction of renal disease prognosis. The applications of CNNs in renal pathology include glomerular counting [4–8], global glomerulosclerosis [9–14], podocyte morphometric analysis [14–17], the classification of diabetic glomerulosclerosis [18], IgA nephropathy [19, 20], glomerular hypercellularity [21], several glomerular changes [22], kidney transplant pathology [23–25], interstitial fibrosis and tubular atrophy [10, 11, 14, 26–28], vascular detection [28], immunofluorescence staining patterns [29], and the classification of normal and abnormal structures in the renal cortex [4, 30–32] (Table 1). However, studies on the development of CNNs that can be successfully applied in the classification of normal and abnormal renal tubules [4, 5, 11, 30], which remains a challenging domain even among renal pathologists, are scarce. Because tubulointerstitial abnormalities significantly predict the outcome of various renal diseases, including acute tubulointerstitial nephritis, diabetic nephropathy, lupus nephritis, and allograft kidneys [33–37], it is crucial to evaluate tubulointerstitial abnormalities quantitatively.

In this study, we aimed to classify normal and abnormal renal tubules precisely by developing a segmentation model using U-Net [38], which is a representative CNN-based architecture mainly used for the segmentation of biomedical images. We improved U-Net by implementing fine finetuning and Dice cross-entropy [39, 40]. We annotated the abnormal tubules in detail, including the atrophic and degenerated tubules as well as tubulitis. The automated classification of renal tubules could help renal pathologists evaluate renal biopsy specimens rapidly and accurately.

## Methods

### Renal biopsy specimens

We used formalin-fixed, paraffin-embedded needle-core biopsies obtained from 21 patients (7 patients 1 h after renal transplantation and 14 patients with tubulointerstitial nephritis) who underwent renal biopsy between 2000 and 2020 at Kanazawa University Hospital and its affiliated hospitals. Because various kidney diseases can involve glomeruli in addition to tubulointerstitial compartments, we needed to collect homogenous samples that involved only the

**Table 1. Deep learning methodologies used for renal pathological studies.**

| Methodology | Stains | Histological primitive | Number of WSIs or cases | Task | Ref. No. |
|---|---|---|---|---|---|
| U-Net with ResNet34 backbone | PAS (paraffin sections) | Glomerulosclerosis, tubular atrophy | 83 WSIs from human transplant biopsies | Segmentation and classification of glomerular and tubular structures | [11] |
| | PAS, MT (paraffin sections) | Arteries, interstitial fibrosis | 65 WSIs from human transplant biopsies | Segmentation of kidney blood vessel and fibrosis | [28] |
| U-Net | PAS (paraffin section) | Glomeruli, sclerotic glomeruli, empty Bowman's capsule, proximal tubuli, distal tubuli, atrophic tubuli, undefined tubuli, capsule, arteries | 137 WSIs from 122 human kidney transplant biopsies and 15 human nephrectomy specimens | Segmentation and classification of multiclass for histological primitives | [4] |
| | PAS, HE, PAM, MT | Glomerular tuft, glomeruli, proximal tubules, distal tubules, artery, peritubular capillaries | 459 curated WSIs from 125 human biopsies with minimal change disease | Multiclass segmentation of histological primitives | [32] |
| | PAS (paraffin section) | Glomerular tuft, glomeruli, tubules, arteries, arterial lumina, tubular atrophy, glomerular size, interstitial expansion | 168 WSIs from 16 humans, 41 healthy mice, 75 murine disease models, 30 other species, and 6 others | Multiclass segmentation of histological primitives | [31] |
| U-Net | PAS (paraffin sections) | Glomeruli | 22 WSIs from mouse kidneys | Glomerular segmentation | [8] |
| U-Net and Yolo V2 architecture CNN | CD3, CD4, CD8, CD20, T-bet, GATA3, CD68, CD163 | Interstitial infiltration of inflammatory cells | 22 WSIs from human kidney transplant biopsies | Quantitative assessment of the inflammatory infiltrates | [25] |
| U-Net and Mask R-CNN | PAS (paraffin section) | Interstitial fibrosis, tubular atrophy, interstitial inflammation | 789 WSIs from human kidney transplant biopsies | Compartment or mononuclear leukocyte detection and tissue detection to predict Banff scores (ci, ct, ti) and rejection | [24] |
| U-Net, DenseNet, LSTM-GCNet, 2D V-Net | PAS (paraffin section) | Glomeruli, mesangial hypercellularity | 400 WSIs from human kidney biopsies with IgA nephropathy | Detection of glomerular location, lesion identification, glomeruli decomposition, mesangial hypercellularity score calculation | [20] |
| U-Net and U-Net cycleGAN | WT1, DACH1 | Glomeruli, podocytes | 110 WSIs from human kidney biopsies with ANCA-associated glomerulonephritis | Podocyte morphometrics | [17] |
| VGG16 | HE (frozen and paraffin sections) | Glomeruli, glomerulosclerosis | 149 WSIs (98 frozen and 51 paraffin sections) from human kidney biopsies | Quantification of the percent global glomerulosclerosis | [9] |
| Inceptionv3 | PAS, HE, PAM (paraffin sections) | Normal, antibody-mediated rejection, T-cell mediated rejection, mixed rejection, borderline T-cell mediated rejection, other disease | 5,844 WSIs from human kidney transplant biopsies | Classification of Banff category | [23] |
| | PAS, PAM (paraffin sections) | Glomerulosclerosis, segmental sclerosis, endocapillary proliferation, mesangial matrix accumulation, mesangial cell proliferation, crescent, basement membrane structural changes | 15,888 glomeruli images from 283 human kidney biopsies | Classification of multiple glomerular findings | [22] |
| DeepLab V2 | PAS | Nonsclerotic glomeruli, sclerotic glomeruli, IFTA | 223 WSIs from human kidney biopsies with 148 diabetic nephropathy and 75 allograft kidneys | Detection and quantification of the percentages of glomerulosclerosis and IFTA | [10] |
| | PAS, HE (paraffin sections) | Nonsclerotic glomeruli, globally sclerotic glomeruli, podocyte nuclei, other nuclei, interstitial fibrosis, tubular atrophy | WSIs from mice kidneys and human kidney biopsies | Segmentation of multiclasses of histological primitives | [14] |
| DeepLabv2 ResNet and RNN | PAS (paraffin section) | Nuclear component, PAS-positive component, luminal component | 54 WSIs from human kidney biopsies and 25 WSIs from mice kidneys | Detection and segmentation of glomerular boundaries on WSIs; diabetic nephropathy classification/prediction | [18] |

*(Continued)*

**Table 1.** (Continued)

| Methodology | Stains | Histological primitive | Number of WSIs or cases | Task | Ref. No. |
|---|---|---|---|---|---|
| SegNet and DeepLab v3+ with ResNet backbone | PAS (paraffin section) | Glomerulosclerosis | 26 WSI from donor kidney biopsies | Glomerular detection and classification | [12] |
| DeepLab v3 and pix2pix GAN | PAS, p57, WT1 (paraffin sections) | Podocyte nuclei | 122 WSIs from mice, rat, human kidney specimens | Automatically detection and quantification of podocytes | [16] |
| SegNet-VGG19 and fine-tuned AlexNet | PAS (paraffin section) | Glomerulosclerosis | 47 WSIs from human kidney biopsies | Segmentation and classification of glomeruli | [13] |
| ResNet-101 | Immunofluorescence (frozen section) | Appearance (granular, linear, pseudolinear), distribution (focal, diffuse, segmental, global), location (mesangial, capillary wall), intensity (0–3) | 12,259 images from 2,542 subjects undergoing kidney biopsies | Classification of immune deposits on glomeruli | [29] |
| fine-tuned NASNet | HE (paraffin section) | Unsupervised extracted features | 68 WSIs form human kidney biopsies with IgA nephropathy | Extraction of features associated with clinical parameters; after clustering, multiclass classification of defined clusters to produce scores | [19] |
| CNN and SVM | PAS, HE (paraffin sections) | Endocapillary hypercellularity, mesangial hypercellularity, endoMes (both lesions) hypercellularity, normal glomeruli | 811 images (300 images of normal human glomeruli and 511 images of human glomeruli with hypercellularity) | Classification of glomerular hypercellularity | [21] |
| Google's Inception v3 | MT (paraffin section) | Interstitial fibrosis | 171 WSIs from human kidney biopsies | Prediction of clinical phenotype | [26] |
| | MT (paraffin section) | Glomeruli | 275 WSIs from 171 human kidney biopsies | Glomerular segmentation and classification | [7] |
| glapathnet (FPN) | MT (paraffin section) | Interstitial fibrosis | 67 WSIs from human kidney biopsies | Prediction of the IFTA grade | [27] |
| AlexNet + SVM | PAS (paraffin section) | Glomeruli, mesangial matrix expansion, tubular nuclei, tubular vacuolization | 98 glomeruli from 17 mice kidneys, 500 image patches of tubule structure | Glomerular detection; classification of glomeruli and tubules | [5] |
| Region-based CNN (AlexNet) | MT (paraffin section) | Glomeruli | 87 WSIs from rat kidneys and 6 WSIs from human kidney biopsies | Glomerular localization and detection | [6] |
| Pix2pix GAN | PAS, WT-1 (paraffin sections) | Glomeruli, podocytes | 24 WSIs from 14 mice kidneys | Automated detection of podocytes | [15] |

ANCA, antineutrophil cytoplasmic antibody; CNN, convolutional neural network; FPN, feature pyramid network; GAN, generative adversarial network; GCNet, graph convolutional network; HE, hematoxylin eosin; IFTA, interstitial fibrosis and tubular atrophy; MT, Masson's-trichrome; NASNet: neural architecture search network; PAM, periodic-acid silver methenamine; PAS, periodic-acid Schiff; ResNet, residual network; RNN, recurrent neural network; SVM, support-vector machine; WSI, whole-slide image; WT-1, Wilms tumor-1

tubulointerstitial compartments for annotation. Thus, specimens with tubulointerstitial nephritis without other involvement were used to annotate abnormal tubulointerstitial structures, whereas specimens collected 1 h after renal transplantation were nearly healthy controls to annotate normal kidney structures. In each specimen, a 2 μm section was stained using a periodic-acid Schiff staining reagent.

This study was approved by the Ethical Committee of Kanazawa University (approval No. 2020–178). The ethics committee waived the requirement for obtaining informed consent from the participants because our study design is retrospective and does not involve any further tests or treatments of the participants. In addition, all data were fully anonymized before we accessed them. Further, all participants had access to the detailed information about the

study, including the purpose, subjects, and content, available on our website. All subjects were allowed to withdraw from the study participation using a written form whenever they wanted. All these processes were approved by the Ethical Committee of Kanazawa University.

## Ground truth training and test sets

From 21 kidney specimens, 311 regions were randomly selected, and 500×500 μm² (approximately 1,000×1,000 pixels) images were captured by a human observer. For each image, the corresponding annotation data were generated using the MATLAB Image Labeler (MathWorks, MA). The annotation data included images labeled pixel-by-pixel for each tissue. Two patterns of classes were marked; (1) five classes: "glomeruli," "normal tubules," "abnormal tubules," "arteries," and "interstitium" and (2) eight classes: "glomeruli," "proximal tubules," "distal tubules," "arteries," "tubulitis," "degenerated tubules," "atrophic tubules," and the "interstitium." These are in the palette format of the PNG images.

The annotations were carried out by a nephrologist with sufficient experience in renal pathology (S.H.). Because the number of renal pathologists is still quite small in Japan, nephrologists are trained and practice renal pathology in most facilities. The annotations performed by S.H. were double-checked by another nephrologist with sufficient renal pathology experience (M.K.) to improve the annotation quality. When the two nephrologists had different opinions, they discussed the issue and then annotated after reaching concordance.

All the normal or abnormal glomeruli were labeled as "glomeruli." Thin ascending limbs of Henle, convoluted distal tubules, and cortical collecting ducts were labeled as "distal tubules." The "arteries" included archery arteries, interlobular arteries, and arterioles. Tubules with infiltration of inflammatory cells and without atrophy or degeneration were defined as "tubulitis." The "atrophic tubules" showed narrowing of the tubular lumen owing to atrophy or the wrinkling of the tubular basement membranes, regardless of inflammatory infiltration, without tubular degeneration. The "degenerated tubules" were defined as tubular abnormalities, such as tubular vacuolation, tubular simplification, budding, loss of brush border, and cell detachment, excluding tubular atrophy and tubulitis. All other unlabeled structures were included in the "interstitium" category.

First, the kidney biopsy images were annotated with eight classes as described. Then, the eight classes were recategorized into five classes. "Proximal tubules" and "distal tubules" were recategorized into "normal tubules," whereas "atrophic tubules," "tubulitis," and "degenerated tubules" were recategorized into "abnormal tubules." The total numbers in the annotated training and test sets are listed in Table 2.

## CNN design

We used U-Net for semantic segmentation. U-Net is a model that applies a CNN [38]. Fine-tuning was implemented using the VGG-16 model [41], which was pretrained on the ImageNet dataset, as the U-Net encoder. The model inputs were the image and annotation data, and the output was the label information for each pixel. We compared the segmentation

Table 2. Number of annotations per class used in the training and test sets of U-Net.

| | | Normal tubules | | Abnormal tubules | | | |
|---|---|---|---|---|---|---|---|
| | Glomeruli | Proximal tubules | Distal tubules | Atrophic tubules | Tubulitis | Degenerated tubules | Arteries |
| Train | 141 | 2,798 | 1,877 | 1,465 | 618 | 1,307 | 205 |
| Test | 35 | 700 | 469 | 266 | 155 | 327 | 51 |
| Total | 176 | 3,498 | 2,346 | 1,831 | 773 | 1,634 | 256 |

models FCN, U-Net, PSP-Net, and DeepLab v3 in a preliminary study, and chose U-Net as the most suitable for the present study because it exhibited the highest accuracy and relatively clear segmented images (S1 Table and S1 Fig).

To train the model, we used 80% of the prepared images, which were randomly selected, and the remaining 20% were used to evaluate the model's performance. One image was only used for the training or the test set. The input images for the model were resized to 512×512 pixels. In addition, we standardized the color appearance by the setting of mean (0.485, 0.456, 0.406) and standard deviation (0.229, 0.224, 0.225) as compared to RGB. Data augmentation was performed during the training process to improve the model's generalization performance, even with a limited amount of data. We adjusted contrast and flipped horizontally at a rate of 50% and rotated in a range of -15˚ to +15˚ for each epoch within random ranges. For contrast adjustment, we calculated the average gray color of the input image in grayscale, and then we created an image "a" of that single gray color. Next, we overlaid the input image and image "a," where the alpha value was a numerical value between 0.5 and 1.5. The alpha value signifies the transparency, and the formula for the output image is given as follows: output = image "a" × (1.0—alpha) + input image × alpha. A value of zero signifies a solid gray image, whereas a value of one signifies that the input image remains the same. All these processes were performed using Python functions. The number of epochs was set to 200. Adam was used as the learning rate optimization algorithm, and Dice cross-entropy was used as the loss function. The output of U-Net was the probability of each label per pixel, and the label with the highest probability was assigned as the predicted label for that pixel.

### Assessment of U-Net's performance

The Dice coefficient (DC), score of the similarity between two sets, was used to evaluate the segmentation accuracy. The DC for two sets A and B, which ranges from 0 to 1, is defined as follows: Because the ground truth (A) and the segmentation result (B) are similar, i.e., the model's performance is higher, the DC value becomes larger and closer to one. We calculated the DC for each label. Cross-validation was performed 20 times, and the median DC value was calculated.

### Agreement rate and time comparison between renal pathologists referring to and not referring to U-Net-segmented images

To evaluate the usefulness of our U-Net algorithm, we examined the agreement ratio between two nephrologists with sufficient experience in renal pathology (R.N. and T.Z.), with and without U-Net-segmented images. For this evaluation, we selected another 15 specimens of tubulointerstitial nephritis obtained through renal biopsies between 2000 and 2020 at Kanazawa University Hospital and its affiliated hospitals. We needed to collect homogenous samples that involved only the tubulointerstitial compartments for validation. Thus, patients with tubulointerstitial nephritis without other involvement were used to estimate abnormal tubulointerstitial structures.

In each sample, a 2 μm section was stained using periodic-acid Schiff staining reagent, and we created whole-slide images for U-Net segmentation. Each renal pathologist evaluated all the biopsy specimens twice. The first assessment was performed without the reference of U-Net-segmented images (U-Net- group), and the other assessment was performed with the reference of U-Net-segmented images (U-Net+ group). There was a washout period of at least two weeks between the U-Net- and U-Net+ groups to avoid habituation effects on the samples. The order of evaluation was crossed: U-Net-→U-Net+ group in nine cases and U-Net +→U-Net-group in six cases. In each review, renal pathologists examined the (1) glomerular

count, the (2) Banff t, ct, and ci scores [42], and the (3) percentage of tubulitis, tubular atrophy, degenerated tubules, and interstitial spaces. Each pathologist recorded the total time taken.

### Statistical analysis

Interclass correlation coefficient (ICC) values (2.1) were calculated for the agreement ratio of continuous variables among the renal pathologists. Cohen's κ was calculated for the agreement ratio of categorical variables among the renal pathologists. Non-parametric parameters of the two groups were compared using the Mann-Whitney U test. The areas of output were compared with those of annotations using linear regression analysis, and the coefficients of determination were calculated. The significance level for all the analyses was set at 0.05.

## Results

### Segmentation performance of U-Net for detecting abnormal tubules

First, we performed the semantic segmentation of five classes (glomeruli, normal tubules, abnormal tubules, arteries, and the interstitium) to clarify whether our U-Net can distinguish between normal and abnormal tubules. Representative examples of the ground truth and segmentation masks used in the test set are shown in Fig 1. The multiclass segmentation performance of U-Net was evaluated and calculated using the DCs listed in Table 3. The highest DCs obtained were for the interstitium and glomeruli. Normal and abnormal tubules had middle DCs. A low DC was observed in the arteries. A confusion matrix shows the way in which one class could be misidentified as a different class (Table 4). Normal tubules were often misidentified as the interstitium but not as abnormal tubules, whereas abnormal tubules were often misidentified as normal tubules (19%) or the interstitium (17%). Arteries were mostly misidentified as the interstitium (64%).

### Detection of different types of abnormal tubules using U-Net

Next, we performed the semantic segmentation of eight classes (glomeruli, proximal tubules, distal tubules, atrophied tubules, tubulitis, degenerated tubules, arteries, and the interstitium) to verify whether our U-Net can be used to detect different types of abnormal tubules in detail. Representative examples of the ground truth and segmentation masks used in the test set are shown in Fig 2. The multiclass segmentation performance of the U-Net was evaluated using the DCs listed in Table 3. The highest DCs were obtained from the interstitium and glomeruli as well as from the five classes of semantic segmentation. Proximal tubules, distal tubules, atrophied tubules, and degenerated tubules had intermediate DCs. Arteries and tubulitis had low DCs. In the confusion matrix, proximal tubules were misidentified as the interstitium (13%) or as degenerated tubules (11%) (Table 5). Distal tubules were misidentified as the interstitium (14%). Arteries were mostly misidentified as the interstitium (60%). Tubulitis was misidentified as the interstitium (21%), distal tubules (15%), or degenerated tubules (15%). Degenerated tubules were misidentified as proximal tubules (17%) or the interstitium (16%). Atrophied tubules were misidentified as the interstitium (17%) or as degenerated tubules (10%).

We also quantified the areas of each class using U-Net to determine whether the algorithm could precisely estimate the area of normal and abnormal tubulointerstitial lesions (Fig 3), which directly resulted in a reasonable prediction of renal prognosis. We found a strong correlation between annotations and the segmentation model predictions in the glomeruli, proximal tubules, distal tubules, and the interstitium. Various abnormal tubules, such as tubulitis, degenerated tubules, atrophied tubules, and arteries, were also moderately correlated between annotations and segmentation model predictions.

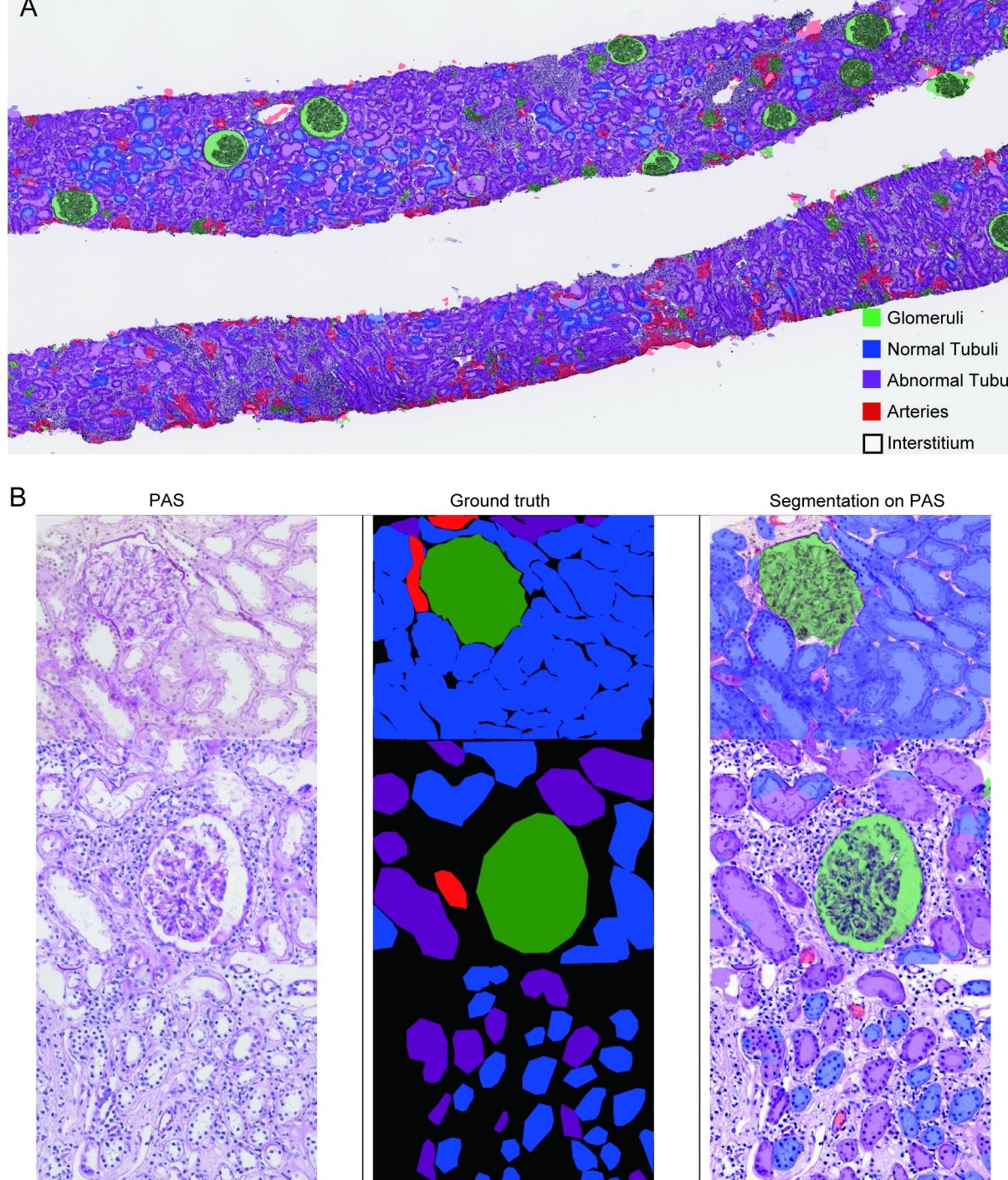

**Fig 1. Representative images of ground truth and eight-class segmentation using U-Net.** (A) Whole-slide image of segmentation using U-Net in a specimen with tubulointerstitial nephritis. (B) PAS-stained slide, ground truth, and segmentation using U-Net. The top row represents a normal specimen; the middle and bottom rows represent specimens with tubulointerstitial nephritis.

**Table 3. Dice coefficients per class.**

| Features | Five classes, median (IQR1, IQR3) | Eight classes, median (IQR1, IQR3) |
|---|---|---|
| Glomeruli | 0.88 (0.55, 0.90) | 0.88 (0.56, 0.90) |
| Normal Tubules | 0.76 (0.64, 0.79) | |
| Proximal Tubules | | 0.69 (0.49, 0.74) |
| Distal Tubules | | 0.65 (0.53, 0.68) |
| Abnormal Tubules | 0.67 (0.56, 0.69) | |
| Atrophied Tubules | | 0.55 (0.38, 0.59) |
| Tubulitis | | 0.30 (0.094, 0.35) |
| Degenerated Tubules | | 0.48 (0.29, 0.54) |
| Arteries | 0.059 (0, 0.16) | 0.027 (0, 0.29) |
| Interstitium | 0.81 (0.74, 0.83) | 0.81 (0.74, 0.82) |

IQR: interquartile range

## Application of U-Net-segmented images to diagnostic situations by renal pathologists

Finally, we evaluated the usefulness of U-Net-segmented images as an aid for routine diagnostic work performed by renal pathologists. We investigated whether referring to five classes of U-Net-segmented images would improve the agreement ratios between two renal pathologists when evaluating tubulointerstitial findings in renal biopsy specimens and the time required for evaluation.

The ICCs for the glomerular count were 0.97 and 0.95 for the U-Net- and U-Net+ groups, respectively (Table 6). The Cohen's κ values of the Banff t, ct, and ci scores were similar at high levels in both groups, ranging from 0.91 to 0.92 in the U-Net- group and 0.81 to 0.94 in the U-Net+ group. The ICCs for the quantitative evaluation of areas in tubulitis, tubular atrophy, degenerated tubules, and the interstitium were low in the U-Net- group (0.14–0.59). However, in the U-Net+ group, the ICCs improved significantly (0.52–0.81), except for degenerated tubules (0.17). Furthermore, referring to the U-Net-segmented images improved the median time for evaluation from 317 s to 214 s [214 s {interquartile range1 (IQR1)180, IQR3 280} in the U-Net+ group vs. 317 s (IQR1 260, IQR3 371) in the U-Net- group; $p = 0.044$].

## Discussion

In this study, we developed a U-Net-based segmentation model to classify the multisystem compartments of renal biopsy specimens primarily related to normal and abnormal tubules. Our developed U-Net could classify normal and abnormal tubules with high accuracy. However, it was still challenging to identify the exact type of abnormal tubules. On the other hand,

**Table 4. Confusion matrix for five-class segmentation using U-Net.**

| | Interstitium | Glomeruli | Normal tubules | Arteries | Abnormal tubules |
|---|---|---|---|---|---|
| Interstitium | **0.81** | 0.0013 | 0.12 | 0.0012 | 0.054 |
| Glomeruli | 0.11 | **0.83** | 0.034 | 0.0048 | 0.022 |
| Normal tubules | 0.12 | 0.0021 | **0.79** | 0.00017 | 0.085 |
| Arteries | 0.64 | 0.096 | 0.063 | **0.096** | 0.10 |
| Abnormal tubules | 0.17 | 0.0039 | 0.19 | 0.0005 | **0.63** |

The ground truth labels are given vertically, and the segmentation model's predictions are given horizontally.

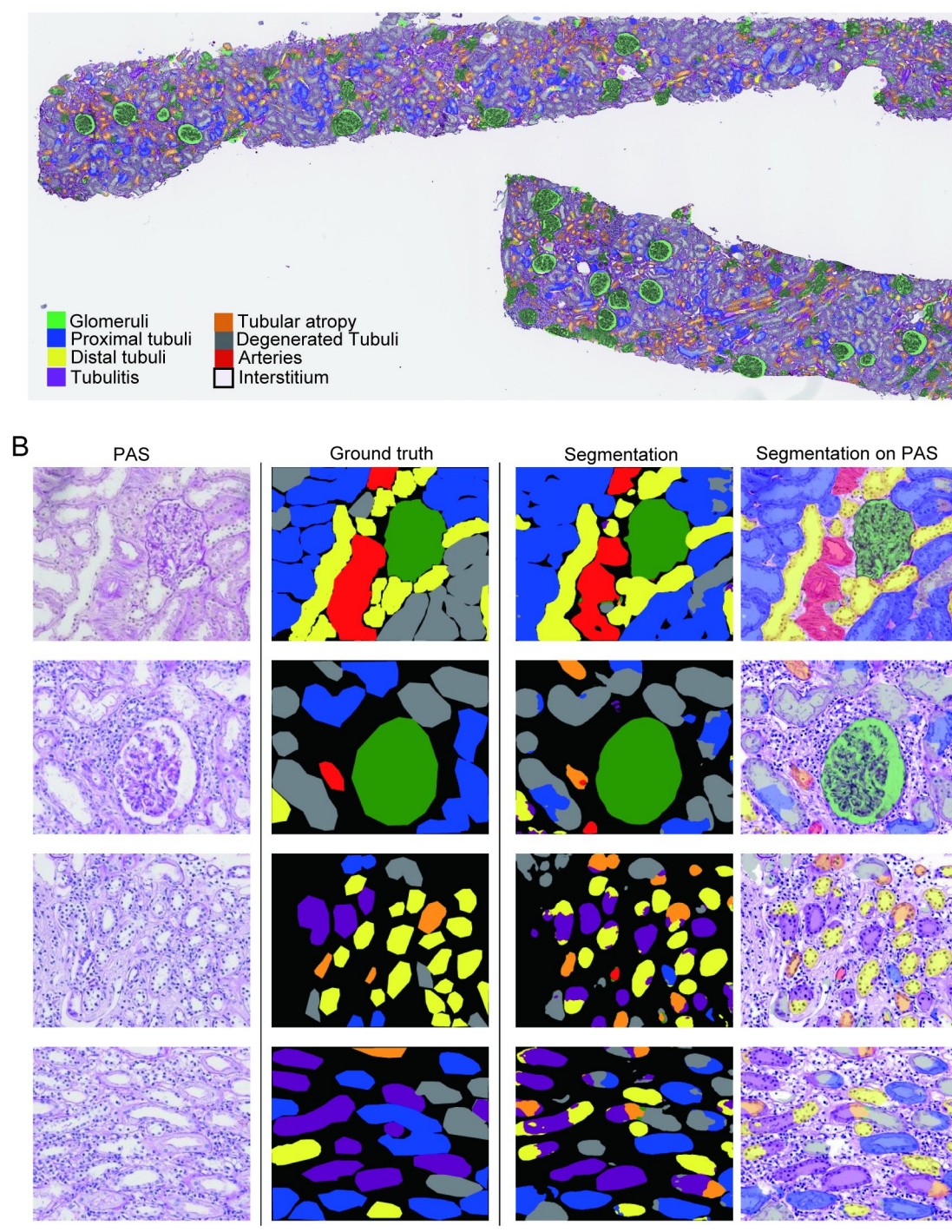

**Fig 2. Representative images of ground truth and eight-class segmentation using U-Net.** (A) Whole-slide image of segmentation using U-Net in a specimen with tubulointerstitial nephritis. (B) PAS-stained slide, ground truth, and segmentation using U-Net. The top row represents a normal specimen, and the second through fourth rows represent specimens with tubulointerstitial nephritis.

**Table 5. Confusion matrix for eight-class segmentation using U-Net.**

|  | Interstitium | Glomeruli | Proximal tubules | Distal tubules | Arteries | Tubulitis | Degenerated tubules | Atrophic tubules |
|---|---|---|---|---|---|---|---|---|
| Interstitium | **0.82** | 0.015 | 0.077 | 0.036 | 0.0024 | 0.011 | 0.029 | 0.014 |
| Glomeruli | 0.083 | **0.85** | 0.030 | 0.013 | 0.0023 | 0.0050 | 0.012 | 0.0053 |
| Proximal tubules | 0.13 | 0.0030 | **0.70** | 0.033 | 0.00067 | 0.017 | 0.11 | 0.011 |
| Distal tubules | 0.14 | 0.0050 | 0.067 | **0.67** | 0.00096 | 0.077 | 0.017 | 0.015 |
| Arteries | 0.60 | 0.086 | 0.020 | 0.036 | **0.14** | 0.023 | 0.039 | 0.064 |
| Tubulitis | 0.21 | 0.00071 | 0.087 | 0.15 | 0.027 | **0.28** | 0.15 | 0.12 |
| Degenerated tubules | 0.16 | 0.0055 | 0.17 | 0.015 | 0.0033 | 0.065 | **0.52** | 0.054 |
| Atrophic tubules | 0.17 | 0.0026 | 0.063 | 0.035 | 0.0012 | 0.085 | 0.11 | **0.53** |

The ground truth labels are given vertically, and the segmentation model's predictions are given horizontally.

our U-Net was suitable for the quantitative evaluation of the area in each class and was helpful as an aid for renal pathologists in evaluating tubulointerstitial lesions among renal biopsy specimens.

In this study, we annotated the most significant number of tubular components to discriminate the types of abnormal tubules by adopting U-Net, which is used for the semantic segmentation of kidney histology [4, 17, 24, 31, 32]. Hermsen et al. achieved multiclass segmentation through U-Net, which showed high DCs on multiclass structures, using whole-slide images obtained from multicenter institutions [4]. Normal tubules were detected highly, but the DCs of both atrophic and undefined tubules were low (0.49 and 0.30, respectively) [4]. In this study, we prepared the most significant amount of annotated data for different types of normal and abnormal tubules, and the detection rate of atrophic tubules was improved. Degenerated tubules were moderately detected, but the model's performance in detecting tubulitis was low. This may be as a result of the diversity of abnormal tubular findings and the fact that different types of abnormal tubular findings often coincide within the same tubules.

The second notable point of the present study is that we improved U-Net by implementing finetuning and Dice cross-entropy. For finetuning, we used the VGG-16 model [41], which was pretrained on the ImageNet dataset, as the U-Net encoder. The introduction of finetuning did not change the accuracy but shortened the learning time taken. It needed about 150 epochs without finetuning to maintain high accuracy, whereas approximately 90 epochs were needed with finetuning. In addition, we adapted Dice cross-entropy as a loss function. Dice cross-entropy is a combination of Dice loss and cross-entropy [39, 40]. Dice cross-entropy improved accuracy more than other loss functions such as focal loss and cross-entropy in our preliminary study. We believe that the use of Dice cross-entropy in renal pathological studies is lacking. Recently, studies have been conducted to detect tubulointerstitial abnormalities using various methodologies. Ginley et al. developed a DeepLab v2-based algorithm to assess interstitial fibrosis and tubular atrophy (IFTA) and glomerulosclerosis in native and transplanted kidneys [10]. They achieved the automated detection and quantification of IFTA lesions by setting IFTA collectively without considering each compartment of IFTA. Bouteldja et al. conducted the multiclass segmentation of healthy and five murine disease models using U-Net [31]. They extracted tubular dilation and atrophy by measuring the tubular diameter. Yi et al. constructed a deep learning-based model through the combination of a mask region-based CNN and U-Net algorithms to recognize normal and abnormal tissue compartments in transplant kidneys, including the Banff t, ci, and ct scores [24]. They applied their algorithms to the prediction of graft survival. Furthermore, Salvi et al. employed two different U-Nets, denoted TSC and TCC, and obtained excellent performance in tubular segmentation (DC = 0.92) [11].

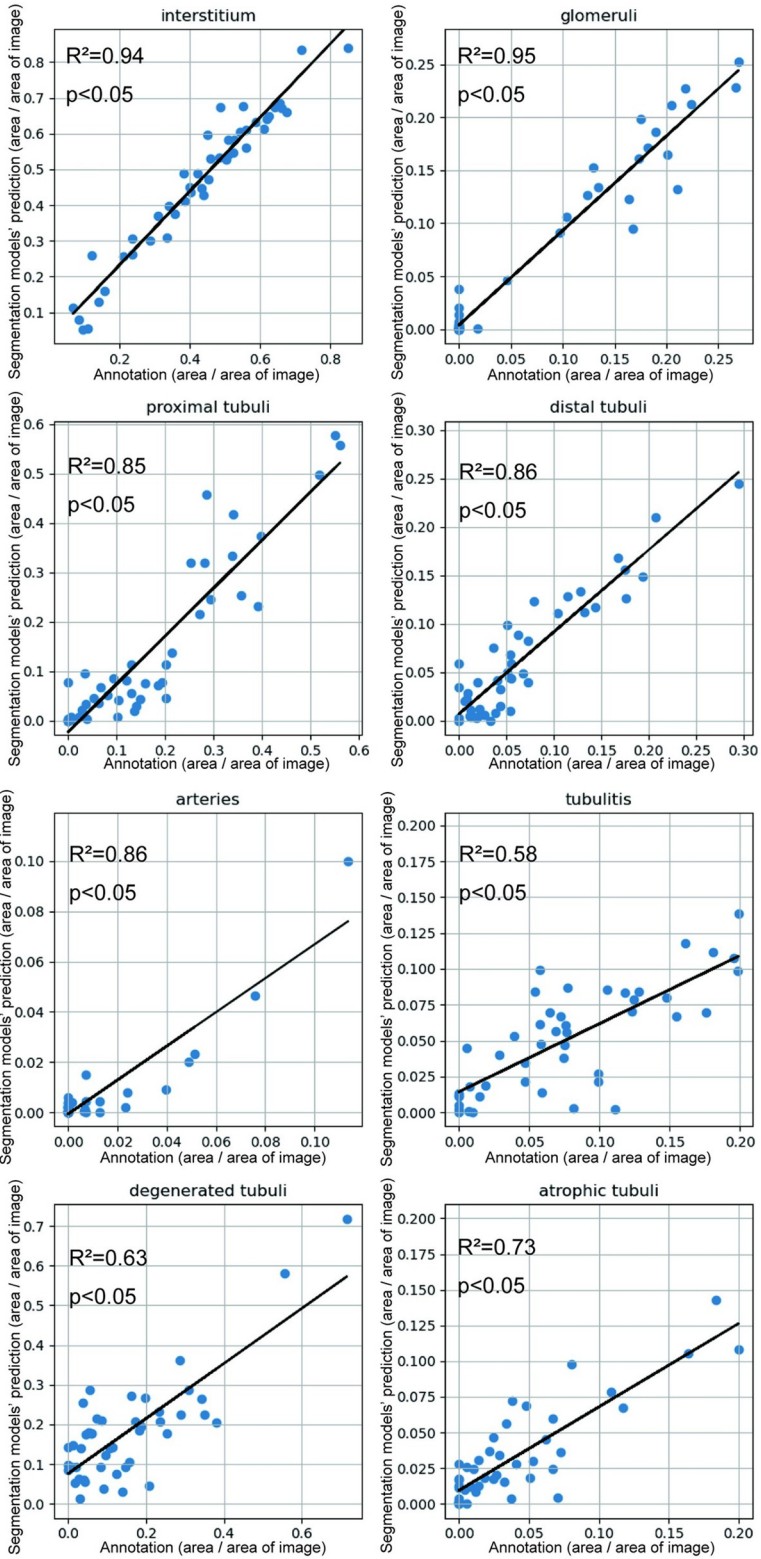

**Fig 3. Correlation of areas between annotations and segmentation model predictions.** There were high correlations in the interstitium, glomeruli, proximal tubules, and distal tubules. Tubulitis, degenerated tubules, atrophied tubules, and arteries were moderately correlated between annotations and segmentation model predictions.

**Table 6. Agreement ratios between renal pathologists with and without U-Net-segmented images.**

| | U-Net- group | | U-Net+ group | |
| --- | --- | --- | --- | --- |
| | κ | ICC | κ | ICC |
| Glomerular count | − | 0.97 | − | 0.95 |
| t score | 0.92 | − | 0.90 | − |
| ct score | 0.91 | − | 0.95 | − |
| ci score | 0.91 | − | 0.82 | − |
| %Tubulitis | − | 0.14 | − | 0.52 |
| %Tubular atrophy | − | 0.28 | − | 0.76 |
| %Degenerative tubules | − | 0.18 | − | 0.17 |
| %Interstitial space | − | 0.59 | − | 0.81 |

ICC, intraclass correlation coefficient

Essentially, although it is still challenging to determine the types of abnormal tubules using U-Net alone, in addition to increasing the validity and the number of annotations, the improvement of deep learning-based methods and their combination with clinical information would be required to improve accuracy in the detection of different types of abnormal tubules and enhancing its significance in clinical outcomes.

Another noteworthy aspect of this study is that referring to the U-Net-segmented images can help renal pathologists in evaluating tubulointerstitial lesions accurately and rapidly. The five-class segmented images were visually easier to understand and more accurate than those of the eight-class segmentation. Therefore, the five-class segmentation images were used to assist renal pathologists in evaluating renal biopsy specimens. The glomerular count and tubulointerstitial compartments of Banff scoring showed the highest agreement with and without U-Net-segmented images. However, interestingly, in the quantitative evaluation of tubular abnormalities, which are more difficult for renal pathologists to assess, U-Net significantly improved the interpathologist agreement ratios, except for degenerated tubules. This may be as a result of the high correlation between the U-Net-segmented and annotated regions in each class. Because abnormal tubulointerstitial areas are associated with worsening renal prognoses in various kidney diseases [26, 33–37], the accurate assessment and quantification of odd tubular areas would improve the quality of the prediction of renal prognosis. Furthermore, the improvement in the time required for evaluation by referring to the segmented images using U-Net is another advantage of U-Net in the reduction of the physical burden on renal pathologists [10]. This includes the development of an application for automated detection and quantification, which would help renal pathologists estimate renal prognosis promptly. In addition, the link between U-Net-based segmentation and clinical information would be useful to predict renal prognosis more precisely. This would notably improve the estimation of renal prognosis compared with the current method of semi-quantification of tubulointerstitial compartments in both native kidney specimens [43] and the Banff-grading system of kidney allografts [42].

This study has several limitations. First, our developed U-Net did not recognize tubules as single structures, and different normal and abnormal tubules were mixed within a single tubule, thereby resulting in lower DCs. Second, a relatively small number of renal pathologists participated in this study to validate the usefulness of referring to U-Net-segmented images. Finally, our developed U-Net had a significantly low accuracy for the "arteries" class. The number of annotated arteries was small. Specifically, the number of annotated arteries was 256 of 311 regions taken and 80% of them were used for training and the remaining 20% for

testing. This is insufficient for U-Net to train for detecting arteries in the test set. In addition, the size of the arteries was extremely small compared with other compartments. The areas of "arteries" are approximately one-fortieth of those of "interstitium." Thus, "arteries" tended to be misrecognized as "interstitium." This study focused on tubulointerstitial structures, and further examination is required to scan the entire renal biopsy specimens, including the arteries.

In conclusion, our deep learning algorithm assisted renal pathologists in detecting and quantifying different types of normal and abnormal tubules in renal biopsy specimens. However, because the current algorithm is still insufficient for the automated detection and classification of different types of abnormal tubules, we must improve its predictive accuracy. Nevertheless, our current algorithm can be expected to help renal pathologists evaluate renal biopsy specimens accurately and rapidly, thereby contributing to highly appropriate clinical decisions.

## Supporting information

**S1 Fig. Representative images of ground truth and eight-class segmentation using various deep learning methods.** PAS-stained slide, ground truth, and segmentation using U-Net. The top row represents a normal specimen, and the second through fourth rows represent specimens with tubulointerstitial nephritis.
(TIF)

**S1 Table. Dice coefficients of various deep learning methods.**
(DOCX)

**S1 File. Dataset of the present study.**
(XLSX)

## Acknowledgments

We would like to thank Yuya Honda and Hiroka Furuya for their support in annotating the images. We would also like to thank Editage (www.editage.com) for English language editing.

## Author Contributions

**Conceptualization:** Satoshi Hara, Emi Haneda, Shigehiro Karashima.

**Data curation:** Satoshi Hara, Emi Haneda, Hidetaka Nambo.

**Formal analysis:** Satoshi Hara, Emi Haneda, Hidetaka Nambo.

**Investigation:** Satoshi Hara, Emi Haneda, Ryo Nishioka, Takeshi Zoshima, Hidetaka Nambo.

**Methodology:** Satoshi Hara, Emi Haneda, Masaki Kawakami, Kento Morita, Ryo Nishioka, Shigehiro Karashima, Hidetaka Nambo.

**Project administration:** Satoshi Hara, Emi Haneda, Shigehiro Karashima, Hidetaka Nambo.

**Resources:** Satoshi Hara, Emi Haneda.

**Software:** Emi Haneda, Masaki Kawakami, Kento Morita, Hidetaka Nambo.

**Supervision:** Mitsuhiro Kawano, Shigehiro Karashima, Hidetaka Nambo.

**Validation:** Emi Haneda, Hidetaka Nambo.

**Visualization:** Satoshi Hara, Emi Haneda.

**Writing – original draft:** Satoshi Hara, Emi Haneda.

**Writing – review & editing:** Masaki Kawakami, Kento Morita, Ryo Nishioka, Takeshi
Zoshima, Mitsuhiro Kometani, Takashi Yoneda, Mitsuhiro Kawano, Shigehiro Karashima,
Hidetaka Nambo.

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
