## [Decision Letter · Decision Letter 0]

6 Apr 2022

PONE-D-22-02424Evaluating tubulointerstitial compartments in renal biopsy specimens using a deep learning-based approach for classifying normal and abnormal tubulesPLOS ONE

Dear Dr. Kawano,

Thank you for submitting your manuscript to PLOS ONE. After careful consideration, we feel that it has merit but does not fully meet PLOS ONE’s publication criteria as it currently stands. Therefore, we invite you to submit a revised version of the manuscript that addresses the points raised during the review process.

We look forward to receiving your revised manuscript.

Kind regards,

Franziska Theilig

Academic Editor

PLOS ONE

Journal Requirements:

2. In the ethics statement in the Methods and online submission information, please ensure that you have specified (1) whether consent was informed and (2) what type you obtained (for instance, written or verbal, and if verbal, how it was documented and witnessed). If your study included minors, state whether you obtained consent from parents or guardians. If the need for consent was waived by the ethics committee, please include this information.

Reviewers' comments:

Reviewer's Responses to Questions

**Comments to the Author**

1. Is the manuscript technically sound, and do the data support the conclusions?

Reviewer #1: Yes

Reviewer #2: Yes

2. Has the statistical analysis been performed appropriately and rigorously? 

Reviewer #1: Yes

Reviewer #2: Yes

3. Have the authors made all data underlying the findings in their manuscript fully available?

Reviewer #1: Yes

Reviewer #2: No

4. Is the manuscript presented in an intelligible fashion and written in standard English?

Reviewer #1: Yes

Reviewer #2: Yes

5. Review Comments to the Author

Reviewer #1: The authors present a deep learning method for the segmentation of renal structures in histopathological images. The manuscript is easy to follow and results are sound. My comments are listed below:

- Background and related works: several papers have been published on this topic (e.g.: doi: 10.1016/j.compmedimag.2021.101930, doi:doi.org/10.3390/electronics9030503, doi: 10.1016/j.cmpb.2019.105273, doi: 10.3390/electronics9101644). authors should at least include these references within the article. To give the reader an idea of current approaches to assessing kidney disease (glomerulosclerosis, tubular atrophy, fibrosis, etc.), the authors could include a table summarizing all state-of-the-art methods.

- Novelty: it is unclear where the novelty lies in the proposed approach, since a well-known segmentation network (UNET) is used for the segmentation task. Was any particular training technique used? Was any kind of pre- or post-processing employed? The authors should highlight the technical novelty (if any) of the work

- Page 7, Line 106: is unclear when five classes are used and when eight classes are employed.

- page 10, Line 140: please specify what kind of operation is performed on RGB images (contrast adjustment)

- Future work?

Reviewer #2: This study sought to distinguish between different kinds of renal tissues on pathology, particularly normal and abnormal tubules using deep learning. To that end, they trained and validated a U-Net based segmentation model. Next, they evaluated the agreement between two pathologists for different tissue types (both with and without the output of the segmentation), as well as the time it took for evaluation.

1) Abstract: Line 47: “whereas the arteries and tubulitis.” Do you mean “arteries and tubules” or do you mean to refer to the pathological condition of “tubulitis?” I presume you mean the latter, but the wording here is a bit confusing when first read, as there appears to be a switch between anatomical structures and a pathological condition.

2) Abstract: Line 49: “The pathological concordance for the glomerular count, Banff t, ct, and ci scores remained high with or without the segmented images.” You may want to clarify if you are referring to the Banff Classification of Renal Allograft Pathology (I presume).

3) Introduction: Line 83: “Because tubulointerstitial abnormalities significantly predict the outcome of renal diseases.” I would consider giving a few examples of these diseases.

4) Methods: Line 95: The Introduction talks about renal diseases in general, but here very specific patients were selected: "We used formalin-fixed, paraffin-embedded needle-core biopsies obtained from 21 patients (7 patients 1 h after renal transplantation and 14 patients with tubulointerstitial nephritis)." It would be helpful to provide an explanation of why these particular patients were selected.

5) Methods, Line 110: “The annotations were carried out by a nephrologist with sufficient experience in renal pathology (S.H.).” Did you consider having more than one nephrologist with renal pathology experience label some of the images to determine their concordance?

6) Methods, Line 133: “We compared the segmentation models FCN, U-Net, PSP-Net, and Deeplab v3 in advance, and we chose U-Net as it was the most suitable for our preliminary data.” Consider citing these other models. Also, please clarify what you mean by “it was the most suitable for our preliminary data.” Did it have the best performance?

7) Methods, Line 135: “To train the model, we used 80% of the prepared images, which were randomly selected, and the remaining 20% were used to evaluate the model’s performance.” Earlier, you state that from 21 kidney specimens, 311 regions were randomly selected. Did regions from the same patient ever end up in both the training set and the test set?

8) Methods, Line 159: “For this evaluation, we selected another 15 specimens of tubulointerstitial nephritis.” Like #4, it would be helpful to have a brief explanation of why this patient population was selected (as opposed to the one referred to earlier).

9) Table 4: Why would you say that the arteries were so frequently identified as interstitium?

10) Results: Line 230, Line 231, Line 234, Figure 3: Please clarify what you mean by “renal outcome” and “output.” Also, in Figure 3, please consider labeling the y-axis with units.

6. PLOS authors have the option to publish the peer review history of their article (what does this mean?). If published, this will include your full peer review and any attached files.

Reviewer #1: **Yes: **Massimo Salvi

Reviewer #2: No

---

## [Author Response · Author response to Decision Letter 0]

20 May 2022

Journal Requirements:

Response: 

We appreciate your comment. Accordingly, we have double-checked and confirmed that our revised manuscript meets PLOS ONE’s style requirements. We have also ensured that all our submission files are named according to the PLOS ONE file naming requirements.

2. In the ethics statement in the Methods and online submission information, please ensure that you have specified (1) whether consent was informed and (2) what type you obtained (for instance, written or verbal, and if verbal, how it was documented and witnessed). If your study included minors, state whether you obtained consent from parents or guardians. If the need for consent was waived by the ethics committee, please include this information.

Response: 

This study was approved by the Ethical Committee of Kanazawa University (approval No. 2020-178). The ethics committee waived the requirement for obtaining informed consent from the participants because our study design is retrospective and does not involve any further tests or treatments of the participants. In addition, all data were fully anonymized before we accessed them. Further, all participants have access to detailed information about the study, including the purpose, subjects, and content, which is available on our website. All subjects were also allowed to withdraw from the study participation using a written form whenever they wanted. All these processes were approved by the Ethical Committee of Kanazawa University. We included these explanations in the Methods section (lines 117–124) and online submission information.

Response: 

We apologize for writing that statement; in fact, we had no funders for the present study. Thus, we would like to amend the statement to “The authors received no specific funding for this work.” We included it in the cover letter as well.

Response: 

We appreciate your comment. We have added our dataset as S1 File to improve the transparency of the present study. 

Reviewers' comments:

Reviewer's Responses to Questions

Comments to the Author

1. Is the manuscript technically sound, and do the data support the conclusions?

Reviewer #1: Yes

Reviewer #2: Yes

2. Has the statistical analysis been performed appropriately and rigorously?

Reviewer #1: Yes

Reviewer #2: Yes

3. Have the authors made all data underlying the findings in their manuscript fully available?

Reviewer #1: Yes

Reviewer #2: No

4. Is the manuscript presented in an intelligible fashion and written in standard English?

Reviewer #1: Yes

Reviewer #2: Yes

5. Review Comments to the Author

Reviewer #1: The authors present a deep learning method for the segmentation of renal structures in histopathological images. The manuscript is easy to follow and results are sound. My comments are listed below:

Response: 

We appreciate your detailed review and comments. We have revised our manuscript according to your advice, and our point-by-point responses are provided below.

- Background and related works: several papers have been published on this topic (e.g.: doi: 10.1016/j.compmedimag.2021.101930, doi:doi.org/10.3390/electronics9030503, doi: 10.1016/j.cmpb.2019.105273, doi: 10.3390/electronics9101644). authors should at least include these references within the article. To give the reader an idea of current approaches to assessing kidney disease (glomerulosclerosis, tubular atrophy, fibrosis, etc.), the authors could include a table summarizing all state-of-the-art methods.

Response: 

We appreciate your comment. According to your advice, we have added all these important related works as references (refs.11–13, 28) and discuss them in Introduction and Discussion (lines 75–83, 364–365). In addition, we have added a table summarizing the state-of-the-art methods for assessing renal pathology (Table 1).

- Novelty: it is unclear where the novelty lies in the proposed approach, since a well-known segmentation network (UNET) is used for the segmentation task. Was any particular training technique used? Was any kind of pre- or post-processing employed? The authors should highlight the technical novelty (if any) of the work

Response: 

We appreciate your comment. The technical novelty of the present study includes finetuning and Dice cross-entropy. 

First, finetuning was implemented using the VGG-16 model (ref. 39), which was pretrained on the ImageNet dataset, as the U-Net encoder. The introduction of finetuning did not change the accuracy but shortened the learning time taken. It needed about 150 epochs without finetuning to maintain high accuracy, whereas approximately 90 epochs were needed with finetuning. 

Second, we adapted Dice cross-entropy as a loss function. Dice cross-entropy is a combination of Dice loss and cross-entropy (refs. 39,40). Dice cross-entropy improved accuracy more than other loss functions such as focal loss and cross-entropy in our preliminary study. We believe that that the use of Dice cross-entropy in renal pathological studies is lacking.

With regard to preprocessing, we employed augmentation such as left–right flipping, rotation, and contrast adjustment, image sizing, and standardization of color information. As for post-processing, the output of the model was the probability of each of the labels for each pixel, and the process was to use the highest probability as the predicted label for the model. 

We included the above explanations in the Introduction section (lines 97–98) and Methods section (lines 164–165, 171–185) and highlighted in the Discussion section (lines 343–352).

- Page 7, Line 106: is unclear when five classes are used and when eight classes are employed.

Response: 

We appreciate your comment. First, the kidney biopsy images were annotated with eight classes as described. Then, the eight classes were recategorized into five classes. “Proximal tubules” and “distal tubules” were recategorized into “normal tubules,” whereas “atrophic tubules,” “tubulitis,” and “degenerated tubules” were recategorized into “abnormal tubules.” For analyses, we first used the five-class set and then the eight-class set. 

To show the above more precisely, we have included these explanations in the Methods section (lines 154–158) and Table 2. 

- page 10, Line 140: please specify what kind of operation is performed on RGB images (contrast adjustment)

Response: 

We appreciate your comment. For the input images for the model, we standardized the color appearance by the setting of mean (0.485, 0.456, 0.406) and standard deviation (0.229, 0.224, 0.225) as compared to RGB, in addition to resizing. With regard to augmentation, we adjusted the contrast and flipped horizontally at a rate of 50% and rotated in a range of -15° to +15° for each epoch within random ranges. Specifically for contrast adjustment, we calculated the average gray color of the input image in grayscale, and then created an image “a” of that single gray color. Next, we overlaid the input image and image “a,” where the alpha value is a numerical value between 0.5 and 1.5. The alpha value denotes the transparency, and the formula for the output image is as follows: output = image “a” × (1.0 - alpha) + input image × alpha. A value of zero signifies a solid gray image, whereas a value of one signifies that the input image remains the same. All these processes were performed using Python functions. 

We have included the above explanations in the Methods section (lines 171–185).

- Future work?

Response: 

We appreciate your comment. We are considering two ways to improve our model. 

First, it would be useful to develop an application for automated detection and quantification to support pathologists. In such an application, when pathologists pointed to a tubule on the screen, the application would indicate the type of tubule. Similarly, automated quantification of abnormal tubules would help pathologists estimate renal prognosis promptly. 

Second, the link between U-Net-based segmentation and clinical information would be useful to predict renal prognosis more precisely. This includes information on how different types of tubular abnormalities affect renal prognosis. This would notably improve the estimation of renal prognosis compared with semi-quantification of tubulointerstitial compartments in both native kidney specimens (ref. 43) and the Banff-grading system of kidney allografts (ref. 42). 

We have included these discussions in the Discussion section (lines 387–394).　

Reviewer #2: This study sought to distinguish between different kinds of renal tissues on pathology, particularly normal and abnormal tubules using deep learning. To that end, they trained and validated a U-Net based segmentation model. Next, they evaluated the agreement between two pathologists for different tissue types (both with and without the output of the segmentation), as well as the time it took for evaluation.

Response: 

We appreciate your detailed review and comments. We have revised our manuscript according to your advice, and our point-by-point responses are provided below.

1) Abstract: Line 47: “whereas the arteries and tubulitis.” Do you mean “arteries and tubules” or do you mean to refer to the pathological condition of “tubulitis?” I presume you mean the latter, but the wording here is a bit confusing when first read, as there appears to be a switch between anatomical structures and a pathological condition.

Response: 

We appreciate your comment. As you have pointed out, our intention was the latter. Thus, we changed the expression from “tubulitis” to “inflamed tubules” in the Abstract (line 48).

2) Abstract: Line 49: “The pathological concordance for the glomerular count, Banff t, ct, and ci scores remained high with or without the segmented images.” You may want to clarify if you are referring to the Banff Classification of Renal Allograft Pathology (I presume).

Response: 

We appreciate your comment. According to your advice, we have clarified this term as t, ct, and ci scores of the Banff classification of renal allograft pathology in the Abstract (lines 50–51). 

3) Introduction: Line 83: “Because tubulointerstitial abnormalities significantly predict the outcome of renal diseases.” I would consider giving a few examples of these diseases.

Response: 

We appreciate your comment. Accordingly, we have given some examples of various renal diseases, such as acute tubulointerstitial nephritis, diabetic nephropathy, lupus nephritis, and allograft kidneys (refs. 34–37), in which tubulointerstitial involvement affects renal prognosis, in the Introduction section (lines 85–87). 

4) Methods: Line 95: The Introduction talks about renal diseases in general, but here very specific patients were selected: "We used formalin-fixed, paraffin-embedded needle-core biopsies obtained from 21 patients (7 patients 1 h after renal transplantation and 14 patients with tubulointerstitial nephritis)." It would be helpful to provide an explanation of why these particular patients were selected.

Response: 

We appreciate your comment. Because various kidney diseases can involve glomeruli in addition to tubulointerstitial compartments, we needed to collect homogenous samples that involved only the tubulointerstitial compartments for annotation. Thus, specimens with tubulointerstitial nephritis without other involvement were used to annotate abnormal tubulointerstitial structures, whereas specimens collected 1 h after renal transplantation were almost healthy controls to annotate normal kidney structures. 

We have included these explanations in the Methods section (lines 108–114). 

5) Methods, Line 110: “The annotations were carried out by a nephrologist with sufficient experience in renal pathology (S.H.).” Did you consider having more than one nephrologist with renal pathology experience label some of the images to determine their concordance?

Response: 

We apologize for giving insufficient explanation and stating that only one nephrologist with renal pathology experience performed the annotation. The annotator (S.H.) was double-checked by another nephrologist with sufficient renal pathology experience (M.K.) to improve the annotation quality. When both nephrologists had different opinions, they discussed the issue and then annotated after reaching concordance. 

We have added these explanations in the Methods section (lines 138–142).

6) Methods, Line 133: “We compared the segmentation models FCN, U-Net, PSP-Net, and Deeplab v3 in advance, and we chose U-Net as it was the most suitable for our preliminary data.” Consider citing these other models. Also, please clarify what you mean by “it was the most suitable for our preliminary data.” Did it have the best performance?

Response: 

We appreciate your comment. We compared the performance of various segmentation models as a preliminary study using 229 images. Of those, 183 were used for training and the remaining 46 were used for testing. In that experiment, U-Net exhibited the highest accuracy in the overall average Dice coefficients and relatively clear segmented images. Thus, we considered that U-Net was the most suitable model for the present study. 

We have added these explanations in the Methods section and added the data as S1 Table and S1 Fig (lines 167–169). 

7) Methods, Line 135: “To train the model, we used 80% of the prepared images, which were randomly selected, and the remaining 20% were used to evaluate the model’s performance.” Earlier, you state that from 21 kidney specimens, 311 regions were randomly selected. Did regions from the same patient ever end up in both the training set and the test set?

Response: 

We appreciate your comment. Here, 311 regions were divided into training and test sets. The 80% of 311 regions were randomly included in the training set and the remaining 20% of 311 regions were used for the test set. As a result, the same patient data may have been used in both the training and test sets, but the same region was used for only the training set or the test set.

 We have added these explanations in the Methods section (lines 171–172).

8) Methods, Line 159: “For this evaluation, we selected another 15 specimens of tubulointerstitial nephritis.” Like #4, it would be helpful to have a brief explanation of why this patient population was selected (as opposed to the one referred to earlier).

Response: 

We appreciate your comment. The reason for selecting these patients is the same as in #4. We needed to collect homogenous samples that involved only the tubulointerstitial compartments for validation. Thus, patients with tubulointerstitial nephritis without other involvement were used to estimate abnormal tubulointerstitial structures. 

We have included these explanations in the Methods section (lines 206–209).

9) Table 4: Why would you say that the arteries were so frequently identified as interstitium?

Response: 

We appreciate your comment. We consider the following two reasons. 

First, the number of annotated arteries was small. Specifically, the number of annotated arteries was 256 of 311 regions taken and 80% of them were used for training and the remaining 20% were for testing. This is insufficient for U-Net to train for detecting arteries in the test set. 

Secondly, the size of arteries is extremely small compared to other compartments. The areas of arteries were about one-fortieth of those of interstitium. Thus, arteries tended to be misrecognized as interstitium. 

We have included these discussions in the Discussion section (lines 400–406).

10) Results: Line 230, Line 231, Line 234, Figure 3: Please clarify what you mean by “renal outcome” and “output.” Also, in Figure 3, please consider labeling the y-axis with units.

Response: 

We appreciate your comment. We mean “renal outcome” as renal prognosis. “Output” means the segmentation model’s prediction. We have revised these expressions to improve consistency (lines 279–283). In addition, we have labeled the x- and y-axis in Figure 3 with units. The x-axis signifies the ratio of areas of annotation divided by area of image, whereas the y-axis signifies the ratio of areas of segmentation model prediction divided by area of image. We have revised these expressions in Figure 3.

---

## [Decision Letter · Decision Letter 1]

27 Jun 2022

Evaluating tubulointerstitial compartments in renal biopsy specimens using a deep learning-based approach for classifying normal and abnormal tubules

PONE-D-22-02424R1

Dear Dr. Kawano,

We’re pleased to inform you that your manuscript has been judged scientifically suitable for publication and will be formally accepted for publication once it meets all outstanding technical requirements.

Kind regards,

Franziska Theilig

Academic Editor

PLOS ONE

Additional Editor Comments (optional):

Reviewers' comments:

Reviewer's Responses to Questions

**Comments to the Author**

1. If the authors have adequately addressed your comments raised in a previous round of review and you feel that this manuscript is now acceptable for publication, you may indicate that here to bypass the “Comments to the Author” section, enter your conflict of interest statement in the “Confidential to Editor” section, and submit your "Accept" recommendation.

Reviewer #1: All comments have been addressed

Reviewer #2: All comments have been addressed

2. Is the manuscript technically sound, and do the data support the conclusions?

Reviewer #1: Yes

Reviewer #2: Yes

3. Has the statistical analysis been performed appropriately and rigorously? 

Reviewer #1: Yes

Reviewer #2: Yes

4. Have the authors made all data underlying the findings in their manuscript fully available?

Reviewer #1: Yes

Reviewer #2: Yes

5. Is the manuscript presented in an intelligible fashion and written in standard English?

Reviewer #1: Yes

Reviewer #2: Yes

6. Review Comments to the Author

Reviewer #1: The authors addressed all my previous comments. The manuscript greatly improved after revision. The revised manuscript is clear and focused.

Reviewer #2: The authors addressed my concerns. They have clarified several points in the methods, which is quite helpful. The detailed addition of prior studies was particularly useful, as well as additional detail regarding areas of technical novelty, model particulars, and future directions.

7. PLOS authors have the option to publish the peer review history of their article (what does this mean?). If published, this will include your full peer review and any attached files.

Reviewer #1: No

Reviewer #2: **Yes: **Hersh Sagreiya

---

## [Editor Report · Acceptance letter]

1 Jul 2022

PONE-D-22-02424R1 

Evaluating tubulointerstitial compartments in renal biopsy specimens using a deep learning-based approach for classifying normal and abnormal tubules 

Dear Dr. Kawano:

I'm pleased to inform you that your manuscript has been deemed suitable for publication in PLOS ONE. Congratulations! Your manuscript is now with our production department. 

Kind regards, 

on behalf of

Dr. Franziska Theilig 

Academic Editor

PLOS ONE